# Combining a Genetic Algorithm and a Fuzzy System to Optimize User Centricity in Autonomous Vehicle Concept Development

Ferdinand Schockenhoff *, Maximilian Zähringer, Matthias Brönner and Markus Lienkamp

Institute of Automotive Technology, Technical University of Munich, Boltzmannstraße 15, 85748 Garching, Germany; maximilian.zaehringer@tum.de (M.Z.); broenner@ftm.mw.tum.de (M.B.); lienkamp@ftm.mw.tum.de (M.L.)
* Correspondence: schockenhoff@ftm.mw.tum.de; Tel.: +49-89-289-10493

**Abstract:** The megatrends of individualization and sharing will dramatically change our consumer behavior. The needs of a product's users will be central input for its development. Current development processes are not suitable for this product development; thus, we propose a combination of a genetic algorithm and a fuzzy system for user-centered development. We execute our new methodological approach on the example of autonomous vehicle concepts to demonstrate its implementation and functionality. The genetic algorithm minimizes the required number of vehicle concepts to satisfy the mobility needs of a user group, and the fuzzy system transfers user needs into vehicle-related properties, which are currently input for vehicle concept development. To present this method, we use a typical family and their potential mobility behavior. Our method optimizes their minimal number of vehicle concepts to satisfy all mobility needs and derives the properties of the vehicle concepts. By integrating our method into the entire vehicle concept development process, autonomous vehicles can be designed user-centered in the context of the megatrends of individualization and sharing. In summary, our method enables us to derive an optimized number of products for qualitatively described, heterogeneous user needs and determine their product-related properties.

**Keywords:** optimization; genetic algorithm; multi-criteria decision methods; fuzzy logic; future product development; user needs; user centricity; autonomous vehicle concept development

## 1. Introduction

As a result of the two megatrends of individualization/personalization [1–3] and sharing [4,5], consumer behavior and the use of products of developed country inhabitants are currently changing dramatically. This poses a major challenge for the development of future products. First, both trends seem to contradict each other in their principles since they demand individualized products, on the one hand, and products optimized for sharing, on the other hand. Second, this represents a shift away from the current development of mass products for private use. Accordingly, a new methodological approach in product development is necessary to meet these apparent contradictions and challenges.

Both trends have in common that they are not driven by the product itself but by the user needs. Therefore, user needs have a key function in the development of future products. The methodology of design thinking [6] has been dealing with this new approach for a while and has established itself in creative sectors. However, the successful use of design thinking is based on the expertise of the developers. For a systematic and partly automated combination of product development methods, a more profound integration of user needs into existing product development processes is necessary to consider technically complicated products. Thus, the aim must use qualitative and linguistically vaguely defined user needs as input, first, and to minimize the product portfolio despite increased individual-

ization, second. The third aim is to generate product-bounded property descriptions as output for integration into existing product development processes.

Therefore, we suggest a combination of an optimization method, to minimize valid product combinations, and a multicriteria decision-making method to combine qualitative user-need description with a technical product concept description. Thus, we choose a genetic algorithm and fuzzy system whose selection is described in detail in Section 3.

In current literature [7–13], approaches that focus on a combination of genetic algorithms and fuzzy systems are already presented. This literature shows the general possibility of combining the two methods and highlights some potential applications. Nevertheless, their usage for incorporating user needs into technical product development is novel and will be presented within this paper.

One of the major application fields of user-centered product development will be autonomous vehicle (AV) concepts. On the one hand, vehicles are one of the most technical products in people's everyday lives, and on the other hand, road-bound mobility will change dramatically in the coming decades. The megatrends of autonomous driving and sharing will disruptively impact future vehicles [5], ([14], pp. 9–13). Both automation and sharing will impact users' attitudes toward the vehicle. While automation enables the performance of personal and individual secondary activities by eliminating the driving task [15], sharing leads to mobility concepts by users abandoning privately owned vehicles [16].

Therefore, in this paper, we start by explaining the optimization of user centricity of products using a combination of a genetic algorithm and a fuzzy system on the contextual issue of AV concepts (Figure 1).

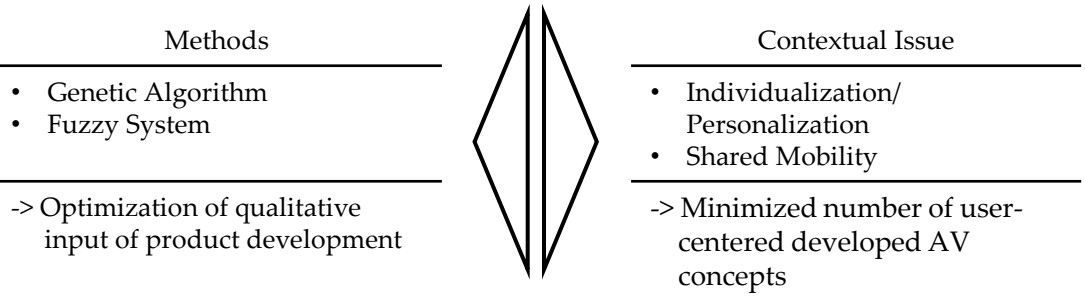

**Figure 1.** Comparison of methods and contextual issue.

In Section 2, we present this contextual issue of AV concept development to introduce the context of our chosen example. In Section 3, based on the analysis of the contextual issue, we derive the usage of the combination of optimization and a multicriteria decision-making method for user-centered product development and our selection of a genetic algorithm and a fuzzy system. Section 4 deals with the model we have developed and demonstrates its functionality using an exemplary use case. Lastly, we present and discuss our conclusions on the findings of this paper.

## 2. Contextual Issue

In this section, we derive the contextual issue of AV concept development as a key application of future product development. Therefore, we present the conventional vehicle concept development to establish the basis of current product development with industrial application. Since personas are the key element of customer centricity of the conventional vehicle concept development, we show their use and limitations. To understand the difference between the current customer centricity and the needed user centricity for the development of AV concepts, we present future mobility solutions. Summing up these considerations, we derive the need for research in the field of product development and in particular in the concept development of AVs. A brief summary of a publication [17] on the concept development process of AVs emphasizes the contextual issue of this paper.

## 2.1. Conventional Vehicle Concept Development

Engineers have been developing conventional vehicle concepts for decades. They use a systematic process that is analyzed in various publications [18–23]. Nicoletti et al. [24] summarize these publications and present the illustration (Figure 2) of the conventional vehicle concept development process.

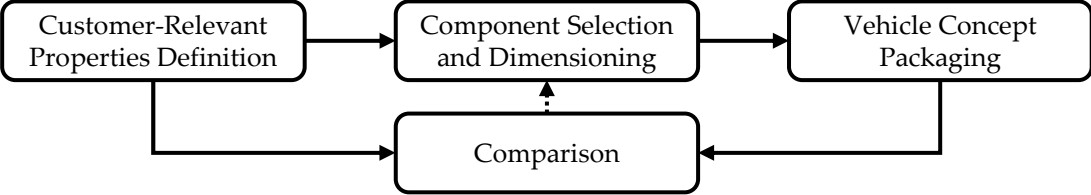

**Figure 2.** Conventional vehicle concept development process, based on [24].

First, customer-relevant properties are defined, representing the customer's view on the vehicle concept. Thus, properties of value for the customer are the input of the conventional vehicle concept development process. These customer-relevant properties are based on an analysis of a potential customer that is a private person who buys, owns, maintains, and uses the vehicle. In general, one vehicle concept should meet all requirements this private person demands [25]. These requirements are usually derived by a persona representing an intended customer.

Personas were established in the 1990s to design customer-centered vehicle concepts [26]. A persona is a hypothetical person who combines the properties and requirements of a group of private people ([27], pp. 61–62), ([28], p. 123)). In addition to the requirements for the vehicle, personality traits and personal context also support customer centricity [26], ([27], pp. 61–62). Finally, the persona represents all personality traits that influence the purchase of a vehicle.

Thus, automotive manufacturers create several personas to represent the group of customers to whom they want to sell their vehicles. In doing so, each persona is usually assigned to one or two vehicles [25]. Consequently, new vehicle concepts are developed with the aim of satisfying the requirements of private persons for the vehicle product.

Using the customer-relevant properties as input, simulations are performed to select and dimension the components of the vehicle concept. For example, longitudinal dynamic simulation is used for the dimensioning of the drivetrain. By packaging components, the vehicle concept is derived. Lastly, iterations are triggered by the comparison of requested customer-relevant properties and properties of the derived vehicle concept. These iterations are necessary due to interdependencies between the different simulations [24].

## 2.2. Mobility Solutions

Driven by the megatrend of sharing [5], ([14], p. 11), new mobility solutions will arise. These mobility solutions will disintegrate the current understanding of the customer in the automotive industry. In addition to vehicle-based mobility providers, numerous other forms of transportation will occur, especially in urban areas. Mobility users will have a choice from a portfolio of bicycles, scooters, rail-based transportation systems, and AV concepts [16]. In our paper, we consider the role of AV concepts in future mobility systems and their development process.

Sharing these AVs leads to mobility as a service (MaaS) models. These are clustered into carsharing and ridesharing. In the former, a passenger shares the vehicle with strangers, whereas in the latter, the ride is shared with one or more strangers who sit directly next to the passenger in the vehicle. In both models, the current notion of a customer from an automotive manufacturer's point of view seems no longer appropriate since passengers are users of a mobility solution rather than a buyer of a specific vehicle [29], pp. 17–21. We, therefore, differentiate between the user of a mobility service and the customer of a specific vehicle concept.

For a user, the focus is therefore not on the requirements of the product but on needs that are to be satisfied by mobility service. Hence, a needs-oriented description of the user for the car-as-a-service system replaces the current description of a persona for the car as a product to achieve user centricity in AV concept development.

### 2.3. Development of Autonomous Vehicle Concepts

In addition to adjustments in the goal of vehicle concept development, the technical change regarding automation must be considered in the development of AV concepts. This affects the steps of component selection and dimensioning and packaging of the vehicle concept. To implement these two disruptive changes, we presented a process for vehicle concept development in a previous publication [17].

In summary, we have included the development of AV functions in the current vehicle concept development process in the technical design by including software components. To achieve user centricity, we add the user-centered mobility needs that consider user groups and their need for mobility. Thus, user groups for designing vehicle fleets substitute personas for designing one specific vehicle. Since we want to enhance the conventional vehicle concept development process (Figure 2), and its sections are commonly known in the automotive sector, we retain the wording of the conventional sections to facilitate the understanding of our AV concept development process.

We refer to the previous publication [17] for the detailed derivation and description of Figure 3.

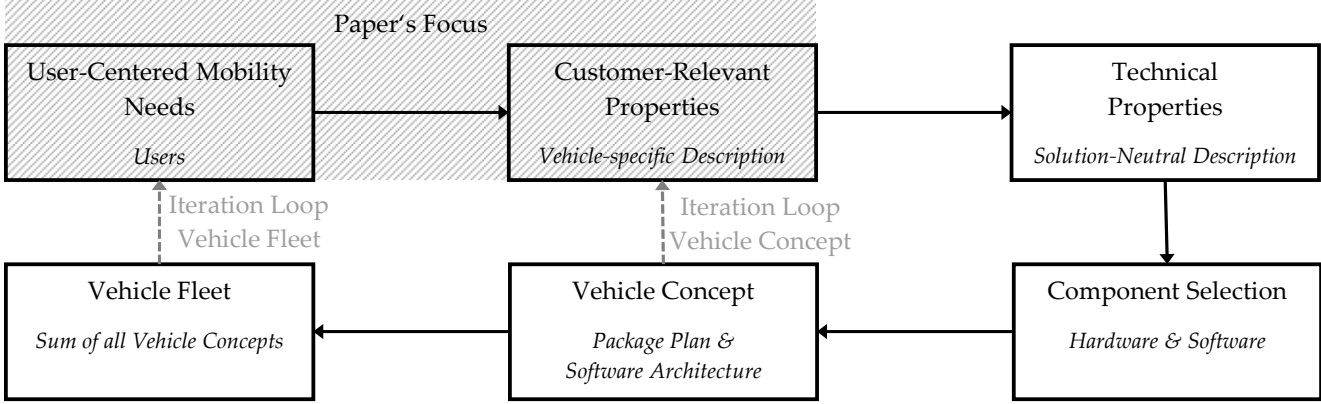

**Figure 3.** Development process of autonomous vehicle concepts, based on [17].

This paper focuses on this new first process section and its derivation of vehicle concepts from mobility needs since this is the novel aspect in product development and requires a new methodological approach. In the next section, we analyze it and provide methods for its implementation.

## 3. Methods

To derive product-related vehicle concept properties from user needs, it is necessary to differentiate the user-centered needs of a mobility system from the customer-relevant properties of a specific vehicle concept (Figure 3). The transformation of user-centered mobility needs into customer-relevant properties to develop the vehicle concepts requires an analysis of this contextual issue to select appropriate methods.

### 3.1. Analysis of the Contextual Issue

The transformation from user-centered mobility needs into customer-relevant properties consists of the following two aspects:

- The number of required vehicle concepts for satisfying user needs must be minimized in order not to design a vehicle concept for every single need;

- The users' mobility needs must be converted into vehicle properties in order to design vehicle concepts for heterogeneous user groups and their mobility needs instead of a customer-oriented persona.

Therefore, we introduce the intermediate module vehicle-bound mobility provision to methodically separate these two aspects [17]. This term describes each individual vehicle concept using the terminology of user needs.

Optimization is therefore performed to minimize the number of required vehicle-bound mobility provisions while specifying a given level of user needs fulfillment. This optimization results in a multidimensional problem whose number of dimensions is unknown, since those depend on the required number of vehicle-bound mobility provisions. An optimization method is needed that independently expands its dimensions until the optimized vehicle-bound mobility provisions satisfy the desired level of user needs fulfillment. This results in the minimum number of vehicle-bound mobility provisions, which leads to a minimum number of vehicle concepts in further processing.

Subsequently, customer-relevant properties are derived from the vehicle-bound mobility provision. This results in a multicriteria decision-making problem since both quantities exist in qualitative form.

Accordingly, the solution to our contextual issue requires a combination of an optimization method and a multicriteria decision-making method (Figure 4). The optimization transfers *n* users with their user-centered mobility needs to *m* vehicle concepts described with the vehicle-bound mobility solution. The multicriteria decision-making method converts each of them into customer-relevant properties of the vehicle concept [17].

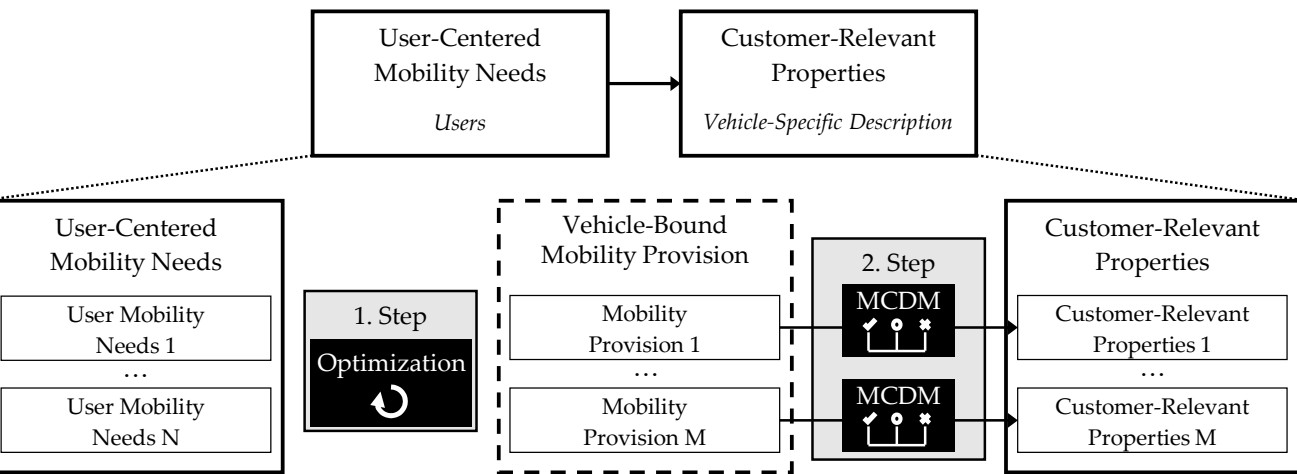

**Figure 4.** Analysis of the contextual issue, based on [17].

In this way—as shown inFigure 4—we solve both contextual issues. The objective of the user-centered approach by replacing individual customer-oriented personas for vehicle purchase with user needs of heterogeneous groups for a mobility offer is enabled by the second step. An upstream optimization as the first step leads to a minimization of the required vehicle concepts; thus, not every single need demands an additional vehicle concept. This added value in the development of AV concepts justifies the additional effort of these two steps.

### 3.2. Method Selection

The following sections first present an overview of possible optimization methods and explain why we have chosen a genetic algorithm for minimizing the required vehicle concepts. Second, we introduce multicriteria decision methods (MCDMs) to select a fuzzy system as the appropriate method to translate the user-centered description to vehicle-specific customer-relevant properties.

### 3.2.1. Optimization

In general, single-objective optimizations are distinguished from multi-objective optimizations. Multi-objective optimizations select decision variables with respect to several objective variables. In the described optimization problem, only one objective variable, the fulfillment of the users' needs, is optimized. Consequently, we deal with a single-objective optimization problem.

According to Coelle et al. [30], such an optimization problem is defined as the minimization or maximization of a function f(x), where $g_i(x) <= 0$ for i = 1, . . . , *m* and $h_j(x) = 0$ for j = 1, . . . , *p* must be satisfied. The global optimum $x_k$ minimizes or maximizes the scalar $f(x_k)$, where $x_k$ is the n-dimensional decision vector of the solution space. The functions $g_i(x)$ and $h_j(x)$ represent constraints. The solution $x_k$ must satisfy these constraints. For example, the constraints can restrict the solution space and guarantee the physical feasibility of a solution. Coelle et al. [30] distinguish three solution strategies for global optimization problems. Enumerative methods are the simplest option. Within a defined search space, each possible solution to the problem is considered, and the best one is determined by comparison. However, these types of methods are unsuitable for large multidimensional search space [30]. Deterministic methods use problem-specific knowledge to control the search in the solution space.

Most real-world problems involve discontinuities and many local extrema. In addition, problem-specific knowledge is usually not available. For these types of irregular problems, stochastic methods are suitable. These methods do not guarantee an optimal solution. However, in the best case, they converge toward the global optimum. Most methods used are evolutionary algorithms, which are based on the processes of natural evolution.

The optimization problem of the first step for deriving customer-relevant properties of AV concepts is directly dependent on the number of vehicles. This in turn depends indirectly on the desired user fulfillment. The higher this value is chosen, the more vehicles have to be provided, and thus, the higher the dimensionality of the problem. Therefore, we propose the use of a stochastic method as a suitable procedure since the dimensionality varies from application to application. Nevertheless, the problem can be classified as highly dimensional. We use the most common procedure, i.e., genetic optimization. An introduction is given in the next section.

### 3.2.2. Genetic Algorithm

Generally, the problem domain is separated from the algorithm domain in evolutionary algorithms. The linkage is performed by a mapping function Γ. The genetic algorithm works with a population of individuals a. In binary notation, an individual is divided into n segments with L bits such that each segment corresponds to a decision variable $x_j$ of the decision vector x. The optimization problem is defined by an objective function F(x) whose equivalent is the fitness function $\phi(a)$ of genetic optimization. The fitness function measures how well an individual solves the optimization problem. The mapping function Γ connects the fitness function with the objective function.

$$\phi(a_i) = F(\Gamma(a_i)) \tag{1}$$

Genetic algorithms consist of five elements that are run through sequentially and iteratively. For the individual elements, several methods exist. In the following, we show a basic variant as used by Nissen [31].

In the first element, an initial population is initialized with μ random individuals $a_i$. The second element evaluates these individuals based on the fitness value $\phi(a_i)$. A selection of individuals for offspring using the fitness value represents the third element. From the population, μ individuals are selected but with laying them back. The probability with which an individual is selected measures the selection probability $p_s(a_i)$ as follows:

$$p_s(a_i) = \frac{\phi(a_i)}{\sum_{j=1}^{\mu} \phi(a_j)} \tag{2}$$

On this basis, a generation of new individuals signifies the fourth element. For this purpose, the elements selection, crossover, mutation, and evaluation are repeated until a complete population is available again. After the selection of two individuals (parents), the crossover probability decides whether a crossover of both strings is performed. Individual bits of a string can be changed by mutation. The evaluation determines whether the newly created individuals replaced previous individuals in the population. Finally, a termination criterion is proved. If this is not met, the elements repeat from the third. If the criterion is met, the solution of the optimization problem is represented by the best individual of the population.

After deriving the use of a genetic algorithm as our optimization method, we address MCDMs and fuzzy logic as the most suitable methods for our contextual issue in the following two sections.

### 3.2.3. Multicriteria Decision Methods

Multicriteria decision problems occur in different application areas, such as economics, finance, management, and development ([32], pp. 1–2). To solve these problems, current literature cites various MCDMs, such as the multi-attribute–utility theory or case-based reasoning [33]. There are also a large number of case-specific adaptations of these methods. To choose an appropriate MCDM for our multicriteria problem, we refer to the study by Verlasquez and Hester [33], which examined the basic multicriteria decision problem methods. This study summarizes fields of application and the advantages and disadvantages of MCDMs. Moreover, we follow the argumentation in Brönner et al. [34], who, based on Verlasquez and Hester [33], have evaluated the MCDM concerning their ability to map cause–effects, the need for exact knowledge of the system, the expandability of the database, and the option of weighting dependencies. They have concluded that a fuzzy logic-based solution best meets these requirements. Additionally, mapping linguistic dependencies are essential for long-term decisions [35]—one of the strengths of fuzzy logic. Following this reasoning, we have chosen fuzzy logic as the basic MCDM to integrate user needs into AV concept development.

### 3.2.4. Fuzzy Logic

A model based on fuzzy logic has the goal to capture the circumstance it is supposed to represent mathematically. In conventional applications, exact input values (also called sharp values) are translated into a linguistic value range (fuzzification) ([36], p. 163). In the following simulation, these linguistic translations are translated into a linguistic value range using IF... THEN... rules (interference) and then translated into an output value (defuzzification) ([32], p. 10), ([37], p. 86). Figure 5 visualizes this described operation of fuzzy systems.

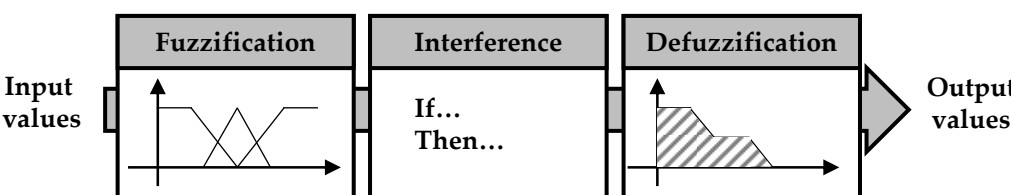

**Figure 5.** Operation of fuzzy systems according to Traeger [37].

These IF... THEN... rules—the major part of a fuzzy system—take the form of Equation (3), where x is the input vector and y is the output vector, represented by the linguistic values $L_j^x$ in the input and $L_j^y$ in the output.

$$\text{If } x = L_j^x \text{ then } y = L_j^y. \tag{3}$$

Thus, within the fuzzification, numerical values, e.g., 100 $\frac{km}{h}$, are translated into linguistic expressions such as "fast." This translation is conducted by so-called membership functions μ, which can represent values between 0 (no membership) and 1 (full membership) ([32] p. 6). These functions are normalized to the value range [0, 1] and modeled as triangular, ramp, or step functions based on expert statements.

To connect fuzzy sets, the linguistic connections "and, not, or" are commonly used ([32] p. 38). These are integrated into the fuzzy system by mathematical operators such as the maximum or minimum operator. A detailed analysis of the selection of these operators is described in Bothe [32], Zimmermann [36], and Traeger [37]. In practical applications, minimum operators are used for the "and" connection and maximum operators for or-connections due to their low computational effort ([37], p. 89). Methods for defuzzification are, for example, the center of gravity method (calculates the center of gravity of the x-coordinate $x_s$).

$$x_s = \frac{\int_{x_A}^{x_B} x \cdot f(x)\, dx}{\int_{x_A}^{x_B} f(x)\, dx} \tag{4}$$

Further methods are the singleton method (uses individual weighted abscissa lines) and the mean-of-maximum method (calculates the abscissa below the mean maximum value) ([32], pp. 12–13), ([36], pp. 234–235), ([37], pp. 104–105]).

## 4. Integration of User-Centered Mobility Needs in Autonomous Vehicle Concept Development

In this section, we want to show how we apply the selected methods genetic algorithm and fuzzy system in our contextual issue. To explain the individual steps, we describe our implementation and present the results for each step using a use case delineated later.

We consider three vehicle types (VTs)—privately owned, carsharing, and ridesharing, as described in Section 2.2. To name them in a commonly used way, we define them as private AV, taxi, and shuttle. This decision is linked to a variety of personal qualitative motives. In any case, an MCDM is required as well. Therefore, we also use a small fuzzy system for this purpose. Thus, the following process runs for every vehicle type.

As discussed in Section 2.2, the description of a user changes significantly when considering AVs. User groups must be considered due to sharing concepts. Second, the transition from driver to passenger is associated with a focus on secondary activities [15]. Here, we distinguish the secondary activities of driving, relaxing, working, sleeping, and load with the fifth secondary activity of load included to also allow luggage and logistic applications. This limitation exists in our implementation. Our methodology allows any number of secondary activities.

The needs for secondary activities become the central parameters $N_p$ (p = 1, ..., 5) for user description. In the case of the first four secondary activities, we call them the key activities. Due to the possibility of ridesharing, the field of privacy and security needs comes to the fore. Therefore, in the user description, with how many users $AMN_p$ (p = 1, ..., 4), we include a secondary activity that may be performed during the ride. The index *p* relates to the secondary activities $N_p$. For the fifth activity load, this consideration is not necessary.

Following the SINUS-Milieus [38] and mobility types as defined by Winterhoff et al. [39], we describe a user by additional characteristics. We consider the willingness for MaaS $C_1$, the willingness to invest $C_2$, and the need for security and safety $C_3$.

Meurle et al. [25] describe a customer within the persona method by his specific use of a vehicle. We consider the essential information contained in this description as relevant in the early concept phase and to be the expression of the driving profile. Therefore, we complement the user description and the characteristics of a driving profile. We define the urban ($DP_1$), rural ($DP_2$), and highway ($DP_3$) parts and the general daily demand ($DP_4$) driving scenarios for this purpose.

We show the two steps from the user description to the vehicle-bound mobility provision toward the customer-relevant properties of AV concepts using a self-defined

exemplary use case. As mentioned above, we need to place a user group, instead of just one typical user at the beginning of the described process.

We assume a family consisting of a married couple with a daughter and a son. We assign the family to the adaptive pragmatic and expeditious milieu of the SINUS-Milieus. These are considered open minded toward new technologies and therefore represent the target group of AVs in the early phase [38]. The father uses the vehicle primarily for riding to work. Accordingly, relaxing and working during this drive is important. For the mother, the fun of driving is in the foreground. For her drive to work, however, a relaxed arrival is also important. Due to the son's leisure activities, the possibility of driving by himself is important; secondary activities play only a subordinate role. The daughter uses a vehicle only for transportation to school. Accordingly, relaxing or sleeping on the way is important for her. To summarize, the characteristics listed in Table 1 result from the milieu affiliation and our fictitious description of the use case on a scale of 0–10.

**Table 1.** Characteristics of the considered use case.

| Characteristics | Variable | Father | Mother | Son | Daughter |
|---|---|---|---|---|---|
| Importance of Self-Driving | N1 | 0 | 8 | 7 | 0 |
| Importance of Relaxing | N2 | 6 | 5 | 4 | 7 |
| Importance of Working | N3 | 7 | 0 | 0 | 0 |
| Importance of Sleeping | N4 | 2 | 4 | 3 | 6 |
| Importance of Load | N5 | 4 | 5 | 5 | 3 |
| Preferred number of passengers while self-driving | AMN1 | 0 | 2 | 4 | 0 |
| Preferred number of passengers while relaxing | AMN2 | 2 | 4 | 4 | 4 |
| Preferred number of passengers while working | AMN3 | 4 | 0 | 0 | 0 |
| Preferred number of passengers while sleeping | AMN4 | 2 | 2 | 2 | 4 |
| Willingness for MaaS | C1 | 7 | 5 | 6 | 8 |
| Willingness to Invest | C2 | 8 | 7 | 6 | 3 |
| Need of Security and Safety | C3 | 4 | 6 | 3 | 7 |
| DP Urban Proportion | DP1 | 6 | 5 | 3 | 5 |
| DP Rural Proportion | DP2 | 2 | 3 | 7 | 3 |
| DP Highway Proportion | DP3 | 5 | 3 | 3 | 0 |
| DP General Need | DP4 | 7 | 5 | 6 | 4 |

A vehicle-bound mobility provision is created for this user group, and the customer-relevant properties of the associated vehicle concepts are derived in the second step.

Methodologically, the two steps subdivided in Section 3.1 are supported by genetic optimization and the fuzzy system. To keep the dimensions of the optimization problem manageable, it first must be clarified by using a small fuzzy system whether a user uses his private AV, a taxi, or shuttle for different mobility needs. After this step, each need for secondary $N_p$ activity is linked to a VT. We know which user chooses which VT (private, taxi, or shuttle) for a mobility need to perform the desired secondary activity $N_p$.

With this knowledge, genetic optimization in the first step of our method is able to determine the required number of vehicles to achieve the desired user fulfillment. We call this proposal and its characteristics vehicle-bound mobility provision. Based on this, we use a fuzzy expert system in the second step to derive the customer-relevant properties of the associated AV concepts. We subsequently show these two main steps in more detail.

### 4.1. From User-Centered Mobility Needs to Vehicle-Bound Mobility Provision

The first step aims to propose the different required derivatives. We call this previous stage of a vehicle concept vehicle-bound mobility provision. We describe this proposal primarily in terms of characteristics of the vehicle interior. This indicates how well various secondary activities are performed in a vehicle. Consequently, a user of the user group is described, on the one hand, by his desire for secondary activities $N_p$ and the number of people $AMN_p$ in this secondary activity. As described, the five secondary activities considered by us are driving, relaxing, working, sleeping, and load $N_p$ (p = 1, ..., 5), and

the three additional personal characteristics are willingness to use MaaS $C_1$, willingness to invest $C_2$, and need for security $C_3$. These three characteristics impact the choice of vehicle design and address the hurdles of sharing from the customer's perspective. Moreover, regarding the customer-valued characteristics, the driving profile DP of a user is decisive.

We describe a derivative proposal j by the global character $GC_{k,j}$ and the global driving profile $GD_{p,j}$ of the users of this proposal, in addition to the characteristic of the vehicle interior $D_{p,j}$ and the number of passenger seats $D_{6,j}$. We use the uniform scale from 0 to 10 for the characteristics, only limiting the number of passenger seats $D_{6,j}$ and the corresponding desire for passengers $AMN_{p,i}$ to nine people. In summary, the presented input and output variables subsequently result in the first step (Figure 6).

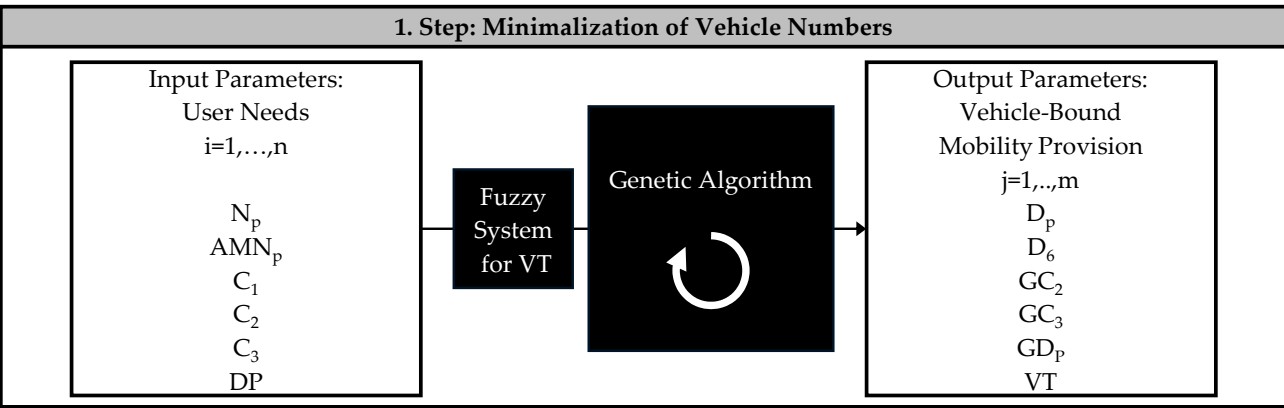

**Figure 6.** First step "user needs to vehicle-bound mobility provision" and its input and output parameters.

Below we describe in detail this first step for linking the input parameters shown with the output parameters.

### 4.1.1. Methodology

As a necessary preliminary element, a fuzzy system is used to link the four key secondary activities $N_1$–$N_4$ with the expected vehicle type VT. As input parameters, this system uses the importance of the secondary activity $N_p$, the number of passengers $AMN_p$, the willingness to MaaS $C_1$, the willingness to invest $C_2$, and the need for safety $C_3$. Qualitative rules are used to determine the choice of vehicle type. The fuzzy system uses 21 rules and represents a necessary preliminary work for genetic optimization. Therefore, we do not show this system in more detail.

Genetic optimization determines the minimum number m of optimal vehicle derivatives based on the n user profiles for a given user fulfillment. From the fuzzy system, we know which secondary activity $N_{p,i}$ of a user i should be satisfied with which vehicle type VT. Therefore, the genetic optimization runs separately for each vehicle expression VT. It is considered that a user uses a private AV and a taxi or shuttle for different trips.

For the mathematical definition of the optimization problem, the decision variables of the vector x must be defined. We use eight decision variables per single vehicle-bound mobility provision; thus, the total number of variables to be optimized is eight times the minimum number of mobility provisions ($n_x = 8\,m$). Six of the eight variables are associated with the vehicle derivative and describe, on the one hand, the quality of the secondary activities $D_{p,j}$ and, on the other hand, the number of passenger seats $D_{6,j}$ of the proposal j = 1, ..., m. The remaining variables of the optimization represent the global investment willingness $GC_{2,j}$ and the global safety need $GC_{3,j}$ of the users of proposal j. The willingness to MaaS of a user is only used for the preliminary element of determining the vehicle type and is therefore not necessary in the optimization. The global driving profile $GDP_j$ of a proposal j based on the driving profiles DP of the users is not part of the optimization due to the strongly increasing dimensionality (+4) but is determined via minimum squared distances according to the users, which are assigned to the considered vehicle.

The decision on the number m of vehicles is made iteratively. Based on the user fulfillment $FF_i$ of each user i by the optimized vehicle-bound mobility provision with m vehicles, the number m is iteratively increased until the following condition is satisfied:

$$\text{mean}_i(FF_i) \geq FF_{wish} \tag{5}$$

The user fulfillment averaged over all users must be greater than or equal to a given user fulfillment $FF_{wish}$. If this condition is not met, the number m is increased by one, and the optimization is restarted. We show the calculation of user fulfillment in detail below.

Calculation of the User Fulfillment

The user fulfillment $FF_i$ of a user by the vehicle-bound mobility provision is composed of three quantities (Figure 7). The fulfillment of the secondary activities $N_{p,i}$ by the quality of the secondary activities $D_{p,j}$ of the flock of derivatives m is measured by $FF_{i,N}$. The match of the number of passenger seats $D_{6,j}$ with the desired number $AMN_{p,i}$ is denoted by $FF_{i,AMN}$. In addition, user fulfillment includes the match of the global characteristics $GC_{2,j}$ and $GC_{3,j}$ with those of the associated users $C_{2,i}$ and $C_{3,i}$. This match is denoted by $FF_{i,C2}$ and $FF_{i,C3}$. In general, it is important to consider whether a derivative j satisfies user i in a secondary activity $N_{p,i}$ in the calculation. Therefore, an assignment of the users to the derivatives proposed by the optimization forms the basis.

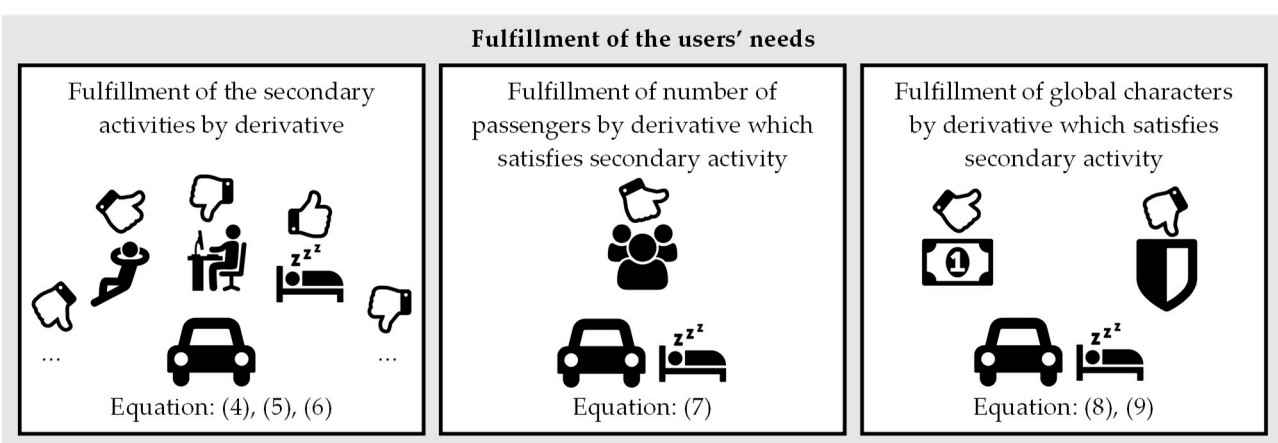

**Figure 7.** Calculation of fulfillment of the users' needs by the derivative.

To compute the individual terms of user fulfillment $FF_i$ of user i, we first show the computation of some auxiliary quantities in matrix notation. The quantity $ff_{i,jp}$ measures how well the wish of secondary activity $N_{p,i}$ of the user i is satisfied through the derivative j with the expression of the interior $D_{p,j}$.

$$ff_{i,jp} = 1 - \frac{f_w(N_{p,i}, D_{p,j})}{N_{p,i}} \mid p = 1, \ldots, 5 \tag{6}$$

To weight under- and over-fulfillments of the characteristics differently, $f_w$ is a weighting function. The quantities $ff_{gc,jk}$ for k = 2,3 is required to calculate the proportions $FF_{i,Ck}$ from the character features. If derivative j best satisfies user *i* in one of the secondary activities $N_{p,i}$ among all m derivatives, the character features $C_{k,i}$ are compared with the global features $GC_{k,j}$ of the derivative.

$$ff_{gc,jk} = \begin{cases} C_{i,k} & \mid \text{if} \quad ff_{i,jp} = \max_j ff_{i,jp} \text{ for } p \in (1, \ldots, 5) \\ 0 & \mid \qquad\qquad\qquad\qquad\qquad\qquad\quad \text{else} \end{cases} \tag{7}$$

Using these quantities, the proportions of user fulfillment $FF_i$ of user i by the vehicle-bound mobility provision, consisting out of m derivatives, can be easily calculated.

The proportion of user fulfillment $FF_{i,N}$ from the secondary activities is calculated as the average over all of the five secondary activities $p$ of the maximum values over the derivatives j since only one derivative has to fulfill the wish $N_{p,i}$.

$$FF_{i,N} = \text{mean}_p\left(\max_j ff_{n,jp}\right) \tag{8}$$

To calculate the proportion $FF_{i,AMN}$, first, the derivative j is determined that best satisfies user *i* in the secondary activity $N_{i,p}$. Its characteristic $D_{6,jpmax}$ is used for the comparison. The fulfillment results as an average value over the four central secondary activities.

$$FF_{i,AMN} = \text{mean}_p\left(1 - \frac{f_w\left(AMN_{p,i}, D_{6,jpmax}\right)}{AMN_{p,i}}\right) \mid p = 1, \dots, 4 \tag{9}$$

The proportion of user fulfillment from the character features $FF_{i,Ck}$ is calculated using the quantity $ff_{gc,jk}$. Therefore, all derivatives j that have an entry in $ff_{gc,jk}$ greater than zero are used.

$$ff_{gc,jk} = 1 - \frac{f_w\left(ff_{gc,jk}, GC_{jk}\right)}{ff_{gc,jk}} \mid ff_{gc,jk} > 0 \tag{10}$$

$$FF_{i,ck} = \text{mean}_j\left(ff_{gcjk}\right) \tag{11}$$

Condition (10) is checked based on the individual proportions of user fulfillment. If this is not fulfilled, the number m is increased by one and the optimization runs again. It has been shown that user fulfillment alone is not sufficient as a fitness function of the proposed genetic optimization. We, therefore, show the setup of these in more detail below.

Calculation of the Fitness Function

The fitness function consists of four parts: the overall fulfillment of the users' secondary needs, the overall fulfillment of the desired number of passengers per secondary activity by the users, the overall fulfillment of users' global character, and a similar fulfillment of the different users' needs (Figure 8).

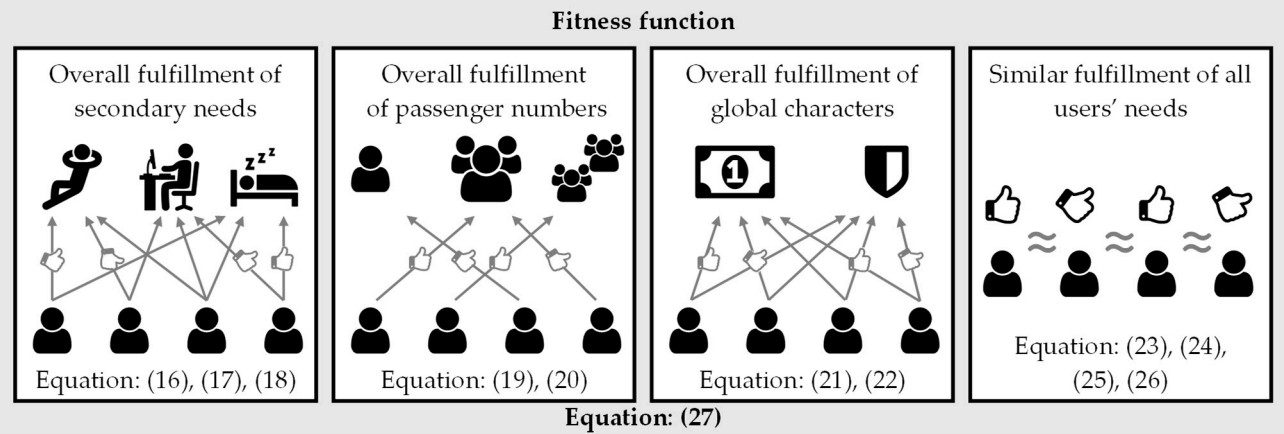

**Figure 8.** Fitness function.

For the calculation of the fitness function, some auxiliary variables are also calculated. The matrices $f_{ip}$ and $ID_{j,ip}$ contain the information which derivative j best satisfies the user *i* in the secondary activity $N_{p,i}$.

$$f_{ip} = \max_j \left( 1 - \frac{f_w(N_{p,i}, D_{p,j})}{N_{p,i}} \right) \text{ for } i = 1, \ldots, n \text{ and } p = 1, \ldots, 5 \tag{12}$$

$$ID_{j,ip} = ID_j(f_{ip}) \mid ID_j \text{ the ID of derivat j} \tag{13}$$

The matrix $AMN_{ip,j}$ stores the wish for additional seat places $AMN_{p,i}$ of user i at the secondary activity $N_{p,i}$ if derivative j best supports secondary activity $N_{p,i}$ among all derivatives m.

$$AMN_{ip,j} = \begin{cases} AMN_{p,i} \mid ff_{i,jp} = \max_j(ff_{i,jp}) \\ 0 \hspace{3cm} \text{else} \end{cases} \tag{14}$$

$$ff_{i,jp} = 1 - \frac{f_w(N_{p,i}, D_{p,j})}{N_{p,i}} \tag{15}$$

The two-character features share in fitness function. In the two matrices $C_{k,ji}$ (k = 2,3), the expression of the character feature $C_{k,i}$ of user i is stored if derivative j best supports user i in one of the secondary activities $N_{p,i}$.

$$C_{k,ji} = \begin{cases} C_{k,i} \mid ff_{i,jp} = \max_j(ff_{i,jp}) \text{ for one p} \\ 0 \hspace{4cm} \text{else} \end{cases} \tag{16}$$

With the help of these quantities, the fitness function is calculated. Thereby, the decision vector x is divided into m vehicle-bound mobility provisions with $\dim(x) = 1 \times 8$ m.

$$x \to D_j = \begin{bmatrix} D_{1,j} & D_{2,j} & D_{3,j} & D_{4,j} & D_{5,j} & D_{6,j} & GC_{2,j} & GC_{3,j} \end{bmatrix} \tag{17}$$

The fitness function of the proposed genetic optimization is composed of four parts. The first part $F_N(x)$ addresses the users' wish for secondary activities and, correspondingly, the offer to exercise them in the derivatives. Therefore, we calculate the deviation $devND_{ip,j}$ between the wish and associated expression of the interior of the derivative. We weigh this by the user's willingness to invest, giving more weight to users with a higher willingness to invest.

$$devND_{ip,j} = \left( f_w(N_{p,i}, D_{p,j})^2 C_{2,i} \right) \forall j \tag{18}$$

Since only one derivative must satisfy the user in the exercise of the secondary activity, we form the product $F_{ip}$ over the derivatives j. If one derivative satisfies the user perfectly, this product disappears, and this part of the fitness function becomes minimal.

$$F_{ip} = \prod_j devND_{ip,j} \tag{19}$$

For the part in the fitness function, the values $F_{ip}$ are summed over all secondary activities p and users i.

$$F_N(x) = \sum_i \sum_P F_{ip} \tag{20}$$

The second part of the fitness function ensures the optimal number of seats $D_{6,j}$ of a derivative j based on the customer's desired $AMN_{p,i}$ depending on the secondary activity. This is carried out by relying on the auxiliary variable $AMN_{ip,j}$, which contains the desired number of passengers $AMN_{p,i}$ of user i if derivative j best supports secondary activity $N_{p,i}$. The maximum number of desired passenger seats placed over all users of derivative j enters the fitness function.

$$AMN_{max,j} = \max_i(AMN_{ip,j}) \tag{21}$$

$$F_{AMN}(x) = \sum_j \left( f_w(AMN_{max,j}, D_{6,j})^2 \right) \tag{22}$$

The third part of the fitness function consists of the character features of the corresponding users. We calculate the deviation between the characteristics of the individual users $C_{k,i}$ and the global characteristics of the vehicle-bound mobility provision $GC_{k,j}$.

$$\text{devC}_{k,ji} = \left( f_w \left( C_{k,ij}, GC_{k,j} \right)^2 \right) \forall\, C_{k,ji} > 0,\ k = 2,3 \tag{23}$$

$$F_{GC}(x) = \sum_k \sum_j \sum_i \text{devC}_{k,ji} \tag{24}$$

We include user fulfillment as the fourth part of the fitness function. We can use this to ensure that users are similarly well satisfied as measured by their willingness to invest. The calculated user fulfillment takes the following form:

$$FF_{user,il} = \begin{bmatrix} FF_{1,N} & FF_{1,AMN} & FF_{1,C2} & FF_{1,C3} \\ \dots & \dots & \dots & \dots \\ FF_{n,N} & FF_{n,AMN} & FF_{n,C2} & FF_{n,C3} \end{bmatrix} \,\Big|\, l = 1,\dots,4 \tag{25}$$

In this part of the fitness function (Equation (25)), we include both the individual terms of user fulfillment and the averaged value.

$$F_{ffs,il} = (f_w(FF_{wish}, FF_{user,il}))C_{2,i} \tag{26}$$

$$F_{ffm,i} = (f_w(FF_{wish}, \text{mean}_l(FF_{user,il})))C_{2,i} \tag{27}$$

The proportion of the fitness function $F_{ff}(x)$ is then calculated by summing over all users i.

$$F_{ff}(x) = \sum_i \left( \sum_l F_{ffs,il} \right) + \sum_i F_{ffm,i} \tag{28}$$

The fitness function $\phi(x)$ of genetic optimization is composed of the four parts shown.

$$\phi(x) = F_N(x) + F_{AMN}(x) + F_{GC}(x) + F_{ff}(x)w_{ff} \tag{29}$$

Via the weight $w_{ff}$, the proportion that leads to a weighted equal fulfillment of the users can be weighted strongly. To ensure the feasibility of the vehicle concepts resulting from the vehicle-bound mobility provision, the solution space must be constrained. For this, we use several constraints.

Introduction of the Constraints

We divide constraints into balanced constraints $C_{eq}(x)$ and unbalanced constraints $C_{ueq}(x)$. As a balanced constraint, we specify that human driving is not possible in an autonomous shuttle, which is assumed due to the expected low price and higher number of passenger seats.

$$C_{eq,1}(x) : D_{1,j} = 0 \,\big|\, VT = \text{Shuttle} \tag{30}$$

For faster convergence of the optimization, we use another balanced constraint that avoids non-essential vehicle interior expression.

$$C_{eq,2}(x) : \sum_i N_{p,i} = 0 \Rightarrow D_{p,j} = 0 \ \forall j \,\big|\, p = 1,\dots,5 \tag{31}$$

In addition, we use three unbalanced constraints that primarily ensure the feasibility of the resulting vehicle concepts. The first unbalanced constraint is to limit the expression of the vehicle interior. It is hardly possible to realize a very high sleeping comfort and at the same time a high expression of the interior space in order to be able to work. Therefore, we limit the sum over the different interior expressions to a maximum of 10. Depending on the requirements of an automotive manufacturer, higher values are also conceivable.

$$C_{ueq,1}(x) : \begin{cases} \sum_{p=1}^{5} D_{p,j} \leq 10 \text{ if } D_{5,j} > 5 \\ \sum_{p=1}^{4} D_{p,j} \leq 10 \qquad \text{else} \end{cases} \tag{32}$$

With regard to the characteristic of the load space $D_{5,j}$ of a derivative, we propose a distinction. If the derivative is primarily optimized for load (logistics), the characteristic is included in the calculation of the sum. If, however, load matters subordinately, the characteristic is excluded from the condition in order to be able to provide a specific amount of luggage space in each vehicle concept if the assigned users express this as a wish.

The second unbalanced constraint $C_{ueq,2}$ also concerns the feasibility of a solution. This constraint ensures that a maximum vehicle interior size $R_{max}(VT)$ is not exceeded due to an excessive number of passenger seats. The maximum available interior size depends on the vehicle type VT and is defined in the three spatial directions,

$$R_{max}(VT) = \begin{bmatrix} R_{max,x}(VT) & R_{max,y}(VT) & R_{max,z}(VT) \end{bmatrix}. \tag{33}$$

The available vehicle interior size is contrasted with the space required to perform the secondary activities. For this purpose, we scale the space requirement per passenger $R_{pp,j}$ via the characteristic $D_{p,j}$ and the global willingness to invest of the users $GC_{2,j}$ of the derivative *j*.

$$R_{pp,j} = \begin{bmatrix} R_{pp,jx} & R_{pp,jy} & R_{pp,jz} \end{bmatrix} \tag{34}$$

We derive the limit values for this from vehicle and aircraft interior dimensions. For each of the three spatial directions, the maximum value is selected via the characteristics $D_{p,j}$. The required space $R_{req,j}$ can be determined via the number of passenger seats and the seat topology. The second unbalanced constraint $C_{ueq,2}$ is then obtained by subtracting the maximum available vehicle interior space.

$$C_{ueq,2}(x) : R_{req,j} - R_{max,j}(VT) \leq 0 \ \forall j \tag{35}$$

We use a third unbalanced constraint $C_{ueq,3}$ to reduce the search space and increase the convergence speed. The minimum values, excluding zero, of the vehicle-bound mobility provision are not to be less than the minimum values of the characteristic expressions of the user group. Similarly, the maximum values of the vehicle-bound mobility provision features are bounded by the maximum values of the user features.

In addition to the eight optimized features (interior space $D_{p,j}$, number of passenger seats $D_{6,j}$, and global character features $GC_{k,j}$), we describe a vehicle-bound mobility provision by the global driving profile. This is obtained as the minimum squared distance to all users assigned to the considered vehicle. We describe the driving profile by the city, rural, and highway fractions, and the level of overall mobility demand. This is also scaled up from 0 to 10.

### 4.1.2. Vehicle-Bound Mobility Provision of the Use Case

For our use case, we set the desired user fulfillment of 75%. On this basis, a fuzzy system and genetic optimization determine the number and characteristics of the vehicle-bound mobility provision. As a result, we obtain five different vehicles for the considered family. Among them, one derivative is expressed as a private AV, three as a taxi, and one as a shuttle. The mother will choose a private AV for fun trips due to her affinity for automobiles, high willingness to invest, and her skepticism toward MaaS. The daughter will use a shuttle for simple trips due to her low willingness to invest. The whole family, however, will favor autonomous taxis with corresponding characteristics for specific trips. Figure 9 shows the vehicle-bound mobility provision based on the user group considered.

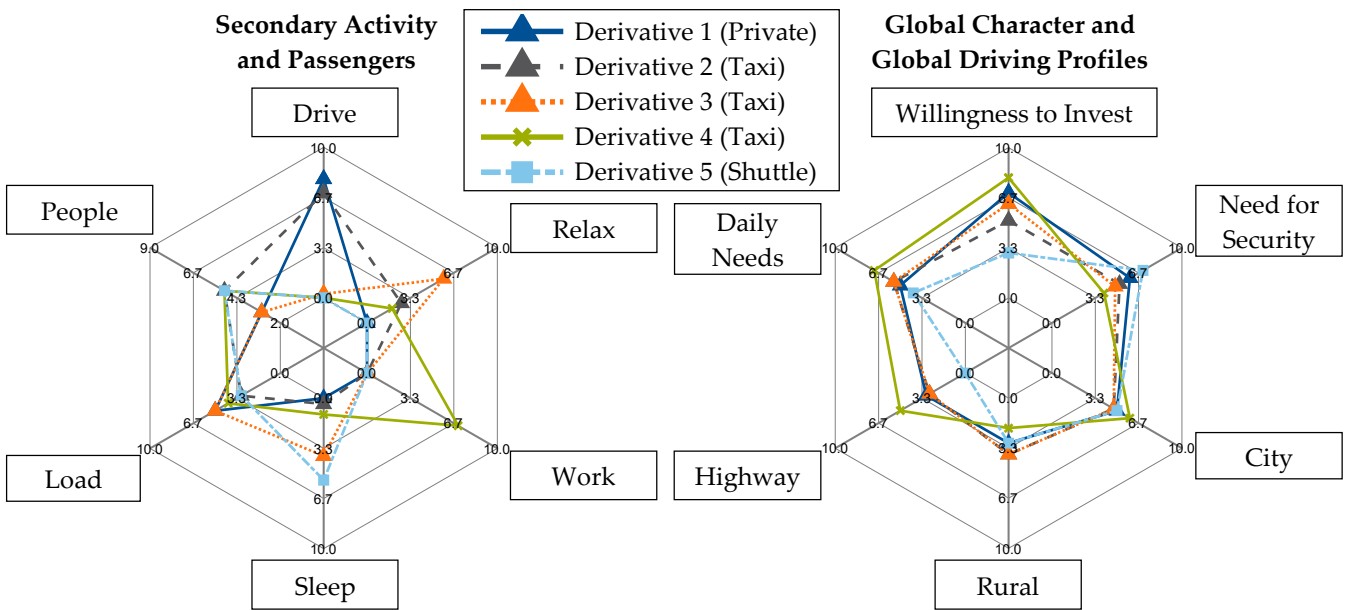

**Figure 9.** Vehicle-bound mobility provision based on the mobility needs of the considered family.

As previously mentioned, Derivative 1 is designed as a private AV and primarily for self-driving. It is striking that Derivative 2 is almost identical to Derivative 1 if we look at Figure 9. This derivative was designed based on the son's wishes. In the case of a family, the son can use the vehicle of the mother (Derivative 1). However, this is not possible for independent user groups, and hence, the solution seems plausible. The third proposal (Derivative 3) fulfills the desire to relax, while the fourth derivative was designed for working during a trip.

The use case also shows that a user uses different vehicles and, conversely, that a vehicle is also used by many different users. Figure 10 illustrates this, showing the possible secondary activities and the seats for people for this secondary activity.

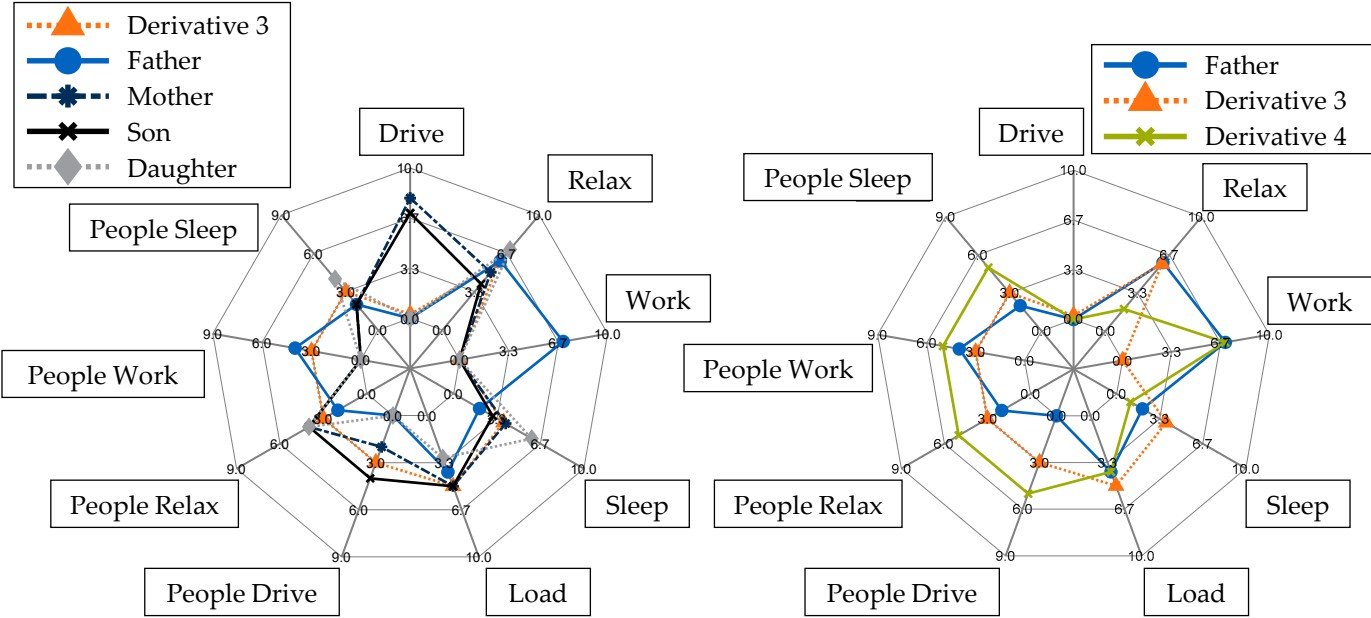

**Figure 10.** Vehicle-bound mobility provision, with the point of view of the Derivative 3 on the left and the father on the right side.

It is evident that two ideal vehicles need to be developed for the father. Depending on his secondary activity, he will either use Derivative 3 or Derivative 4. Derivative 3 illustrates that one derivative is used by several users. The third proposal was designed to meet the needs of all users considered and contribute to their satisfaction.

The vehicle-bound mobility provision represents a design that can be adapted by the concept engineer. This intermediate module is then used to derive the customer-relevant properties for the associated vehicle concepts. We show this in the next section, first in theory and then exemplary for the use case discussed here.

### 4.2. From Vehicle-Bound Mobility Provision to Customer-Relevant Properties

The customer-relevant properties of AV concepts can be derived from user descriptions using the vehicle-bound mobility provision. In the second step, we use a fuzzy expert system to link the characteristics of a vehicle-bound mobility provision with the customer-relevant properties of AV concepts. Figure 11 illustrates this process.

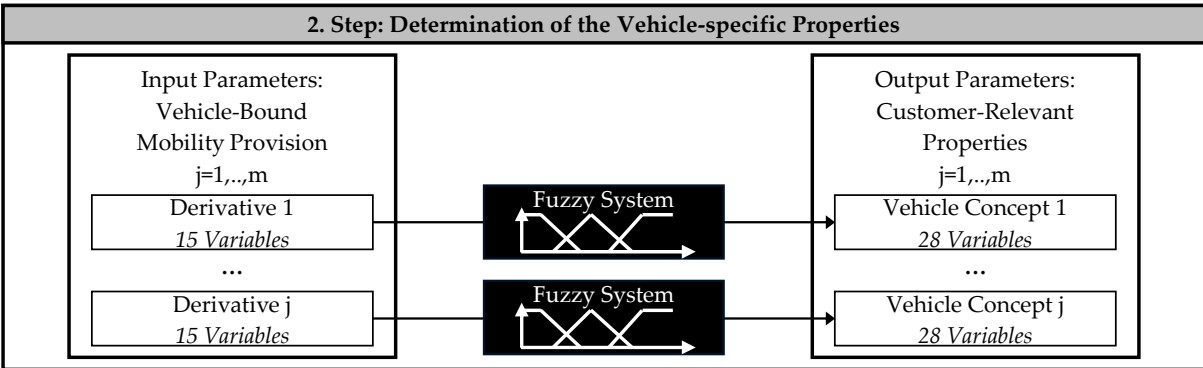

**Figure 11.** The second step of "Vehicle-Bound Mobility Provision to Customer-Relevant Properties" and its input and output parameters.

To detail this process, we first show the theory of the fuzzy expert system and then the application to the exemplary use case.

#### 4.2.1. Methodology

We have defined the input and output variables of the fuzzy expert system. The known features of the vehicle-bound mobility provision served as input variables. For linguistic description, we used three to four fuzzy sets. The input variables are described in Table 2.

**Table 2.** Input parameters of the fuzzy system and its numerous and linguistic range.

| Input Parameter | Numerous Range | Linguistic Range |
|---|---|---|
| expression of driving | [0, 10] | [no expression, high expression] |
| expression of relaxing | [0, 10] | [no expression, high expression] |
| expression of working | [0, 10] | [no expression, high expression] |
| expression of sleeping | [0, 10] | [no expression, high expression] |
| load capacity | [0, 10] | [very low, high] |
| number of passengers | [2, 9] | [low, high] |
| axial space requirement | [0, 10] | [low, high] |
| lateral space requirement | [0, 10] | [low, high] |
| vehicle type | [1, 2, 3] | [Private, Taxi, Shuttle] |
| global invest readiness | [0, 10] | [low, high] |
| global safety readiness | [0, 10] | [low, high] |
| DP city share | [0, 10] | [low, high] |
| DP rural share | [0, 10] | [low, high] |
| DP highway share | [0, 10] | [low, high] |
| DP daily need | [0, 10] | [low, high] |

The output variables represent the customer-relevant properties of an AV concept. Based on the overall vehicle properties defined by Schockenhoff et al. [15], we proposed a total of 28 customer-relevant properties at the specification level by adapting existing customer-relevant properties [23,40,41] and adding new properties through studies and reports ([42], p. 13), ([43], p. 14), ([44], p. 18), ([45], p. 8), ([46], p. 64). These are described inTable 3.

**Table 3.** Output parameters of the Fuzzy-System and its numerous and linguistic value range.

| Output Parameter | Numerous Range | Linguistic Range |
|---|---|---|
| Quality of Axial Dynamics | [0, 10] | [low, high] |
| Quality of Lateral Dynamics | [0, 10] | [low, high] |
| Quality of Vertical Dynamics | [0, 10] | [low, high] |
| Maneuverability | [0, 10] | [low, high] |
| Bad Road Capability | [0, 10] | [low, high] |
| Passive Safety | [0, 10] | [low, high] |
| Luggage Space | [0, 10] | [low, high] |
| Boarding Comfort | [0, 10] | [low, high] |
| Boarding Time | [0, 10] | [low, high] |
| Leg Room | [0, 10] | [low, high] |
| Shoulder Room | [0, 10] | [low, high] |
| Head Room | [0, 10] | [low, high] |
| External Communication | [0, 10] | [standardized, high] |
| Built-in Infotainment | [0, 10] | [purposeful, user individual] |
| Infotainment Individualization | [0, 10] | [low, high] |
| Interior Recognition | [0, 10] | [conforming to approval, high expression] |
| Driving Style: Comfort | [0, 10] | [low, high] |
| Driving Style: Safety | [0, 10] | [low, high] |
| Driving Style: Time Potential | [0, 10] | [low, high] |
| Driving Style: Consumption | [0, 10] | [low, high] |
| Driving Style: Degree of Freedom | [0, 10] | [low, high] |
| Quality Exterior Design | [0, 10] | [low, high] |
| Range | [0, 10] | [low, high] |
| Acoustic Interior | [0, 10] | [tolerable, very silent] |
| Environmental Monitoring | [0, 10] | [drivable, high] |
| Active Safety | [0, 10] | [drivable, high] |
| Costs | [0, 10] | [low, high] |
| Ecology | [0, 10] | [low, high] |

The input and output parameters were fuzzified. For this purpose, the membership functions of the fuzzy set were defined. Following the argumentation of Schröder [47], we used triangular functions for simplified interpretation. Figure 12 shows the fuzzification for an input parameter.

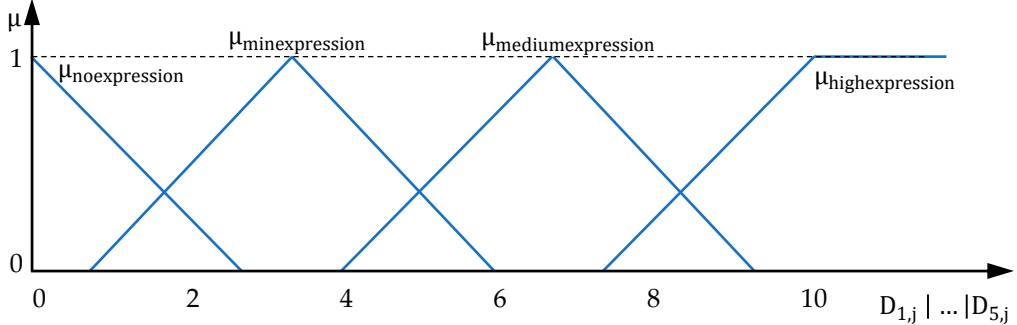

**Figure 12.** Fuzzification of the input parameter Expression of vehicle interior $D_{p,j}$.

The core element of a fuzzy system is its rule base. This rule base links the input parameters with the output parameters in the form of qualitative IF... THEN... relationships. On the one hand, this rule base can be established by a literature review. However, in this study, we propose entirely new, partly unknown characteristics of a vehicle-bound mobility provision and customer-relevant properties of AV concepts.

We have used an expert survey to establish the rule base. In this survey, 14 experts from science and industry participated. All of them work in a specific field of our rules. We asked ergonomic experts for the anthropometric geometrical parameters of the interior and human machine interface (HMI) experts for the parameters coupled to infotainment and entertainment. Moreover, future e-mobility experts were surveyed about the general aspects and vehicle dynamic experts for the vehicle behavior. Most of the experts from the science fields work in the research project UNICARagil [48], in which they build self-driving vehicles. All experts together established 538 rules linking the 14 input parameters to 28 output parameters. On a scale of 0–5, the experts rated their knowledge in the concept development of AVs at 3.92.

For the implementation of the fuzzy expert system, we used the MATLAB Fuzzy Logic Designer. The first task was to choose the type of fuzzy system. The Mamdani system is well suited for human input and is the foundation of many industrial applications [49]. Therefore, we also used the Mamdani system. For rules with multiple premises, the second task was to specify the mathematical operators of the AND and OR relations. We followed Lee [50] and used the product operator for the AND operation and the maximum operator for the OR operation.

If several rules led to the same statement, the different degrees of membership of this one fuzzy set had to be merged. This task is called implication. For fuzzy expert systems, the choice of the product operator was proposed [50]. The aggregation followed the implication. We chose the maximus operator as the operator [51]. The final choice was the defuzzification method. We followed the recommendation of Mathworks [52] and used the most used centroid method.

In the following section, we will show the second step in the development of the customer-relevant properties of AV concepts in a potential application. The input is the vehicle-bound mobility provision shown for the use case described above.

### 4.2.2. Customer-Relevant Properties of the Use Case

The output variables of the fuzzy expert system represent the customer-relevant properties of the AV concept. In the following, as an example, we consider property field dynamics, general and driving style of Derivatives 1, 3, and 5 of the previously optimized vehicle-bound mobility provisions. All customer-relevant properties are expressed on a uniform scale from 0 to 10 due to their solution-neutral formulation. Figure 13 shows the distinct properties of the private AV, the taxi, and the autonomous shuttle.

The customer-relevant properties of dynamics address driving behavior. In the lateral dynamics, the private AV concept for the secondary need driving has a high value whereas the taxi for relaxing and the shuttle for sleeping indicate a low one. The characteristics of the longitudinal dynamics are generally low. The values for lateral dynamics seem plausible, while a higher expression would be expected for longitudinal dynamics of the private AV. For AVs, the quality of the vertical dynamics can generally be classified as higher. The difference between the taxi for relaxing and the shuttle for sleeping seems plausible regarding the secondary activity, although taxis expect a higher level of vertical dynamic than shuttles. The levels of passive safety seem appropriate. The changed seating or lying positions of the passengers due to the secondary activities of relaxing and sleeping require these higher expressions in the taxi and shuttle.

In the general characteristics, the higher expression of the exterior design of the taxi, compared to the shuttle, seems plausible. However, the low salience of the private AV does not match our expectations. The high expression of the environmental monitoring of the taxi and the shuttle is expected due to the permanent autonomous operation. The interior

acoustic is the highest in the shuttle for the secondary activity sleeping and the lowest in the taxi. Regarding an expected ranking of good acoustics for private AV and worse for shuttles and good acoustics for sleeping and worse for driving, this seems a fitting result in its interactions.

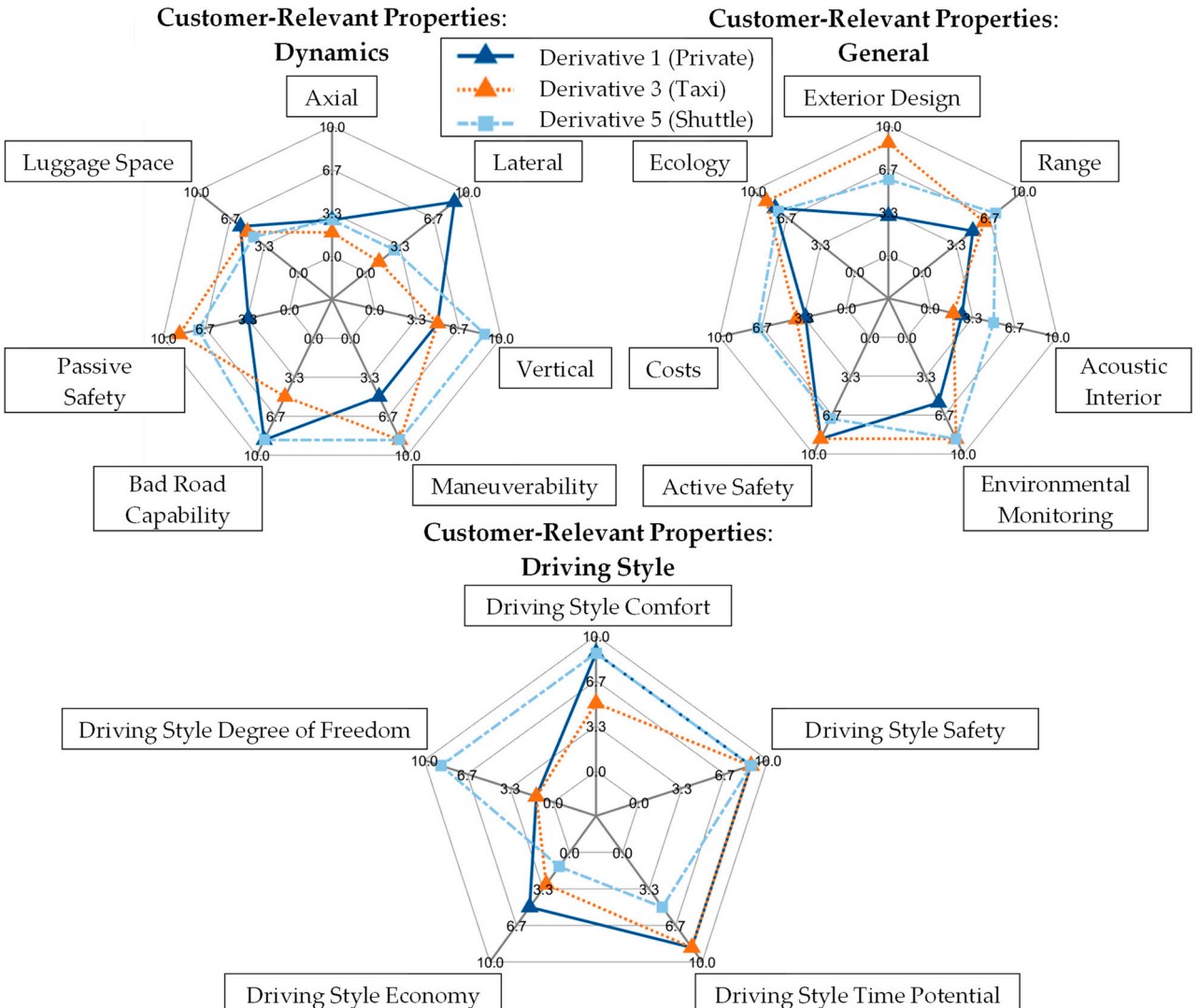

**Figure 13.** Customer-relevant properties for dynamics, general, and driving style properties.

The driving style of the three vehicles also differs. The driving style of the private AV can be neglected since in manual driving the driver specifies it and the shown driving style is therefore exclusively relevant for the subordinate secondary activity of relaxing. The comfort of the driving style of the shuttle is very high due to the intended secondary activity of sleeping, while in terms of time potential, the shuttle has a significantly lower characteristic than the taxi. This is plausible for a ridesharing vehicle. The only questionable property is the very high value of the degree of freedom of the shuttle's driving style. This characterizes the possibility to change the driving style, and a shuttle is unlikely to allow adaptivity of the driving style by the user.

## 5. Discussion

We have shown that future products can be developed based on the needs of their users. Methodologically, this requires the optimization of qualitative, differing input and output variables. While the input variables define user needs, the output variables must

describe the required product-related properties. This methodological challenge is solvable by combining a genetic algorithm and a fuzzy system.

We have presented the method on the example of AV concepts. We have highlighted that future mobility solutions satisfy more than just the need for mobility by offering feasible secondary activities and have shown that these user requirements must become part of the development process of AV concepts. The presented methodology integrates user groups and their needs into the vehicle concept development process. Thus, it is possible to design vehicle concepts for a private person using personas and to also optimize vehicle fleets for user groups. Regarding MaaS offerings with AV fleets, this possibility is essential for user-oriented vehicle concept development.

Thus, the performed example demonstrates that our method works and contributes to the development of future products in the context of individualization/personalization and sharing.

Nevertheless, our implementation has weaknesses. Both fuzzy systems would have to be integrated into the optimization in order to optimize the vehicle-specific output variables. Thus, the optimal vehicle fleet would be directly derivable in vehicle-specific variables for given user needs. However, we opted for a sequential, rather than an integrated, linkage of the process steps. Since we are integrating user needs of autonomous mobility into the vehicle concept development process for the first time, we want to obtain process knowledge with an intermediate module. This is the way we ensure the plausibility of the results. At a later stage in a potential industrial application, the small fuzzy system of the first step should be removed and the selection of the vehicle type should be part of the optimization. Furthermore, the integration of the huge fuzzy system of the second step into the optimization should be examined.

Focusing on the results, the fuzzy system presented and discussed in Section 4.2.2. still do not seem to be completely plausible. We attribute this to the rule base. Since the experts are surveyed about a future topic, different considerations inevitably arise during rule generation ([53], p. 27). Hence, the fuzzy system seems to be the appropriate method since it averages and combines different statements. Nevertheless, expert validation with analytically defined scenarios is necessary to highlight important rules by weighting them [36]. If possible, the number of rules should not be increased to preserve the interpretability of the model and its results [54]. Furthermore, the results are dependent on the required level of fulfillment of the user needs, which is the termination criteria of the optimization. Therefore, our results are only an example and in a potential industrial application, the required level of fulfillment of the user needs must be purposefully chosen.

The key advantage of our approach is integrability into existing product development processes, e.g., vehicle concept development processes. Thus, an existing, well-functioning process is transformed by our method from a customer-oriented to a user-oriented approach. Although adaptations in the individual process are necessary for the development of AVs [17], only adding the presented method enables a user-centered development of AV concepts.

## 6. Conclusions and Outlook

Driven by the megatrends of individualization/personalization and sharing, we have identified the demand for a change from customer-oriented product development for private ownership to a user-centered approach using user group needs for a sharing society. Using the development of AV concepts as an example, we have illustrated this problem.

For this purpose, we designed a method that combines a genetic algorithm and a fuzzy system. On the one hand, this enables the minimization of the required product variants while satisfying the user needs, in our example, the AV concepts in the fleet, and, on the other hand, transfers variables of the user needs into product-related ones. Thus, our methodological approach will support the future user-centered development of products.

This paper includes an exemplary use case of designing a fleet of AVs for a family of four and their heterogeneous mobility needs. This use case illustrates the shift from

conventional vehicle concept design, in which one vehicle must meet all of a customer's needs. All family members use multiple vehicle concepts, mostly via MaaS offerings, to fully satisfy their mobility needs.

With the application to this use case, we demonstrate the functionality for a technically complex, future product represented by AV concepts. The transfer of the methodological approach to other products is still pending but should be easily feasible.

Therefore, further research should focus on adapting the methodological approach to other future products. This will improve the method and underline its strengths.

For our example of AV concepts, the validation of the rule base of the fuzzy system should be performed. Based on this, an overall system validation can be performed to verify the interaction of the genetic algorithm and the fuzzy system.

**Author Contributions:** First author, F.S.; Conceptualization, F.S.; methodology, F.S. and M.Z.; software, M.Z.; investigation, F.S. and M.Z.; data curation, F.S. and M.Z.; writing—original draft preparation, F.S. and M.Z.; writing—review and editing, F.S., M.B. and M.L.; visualization, F.S. and M.Z.; supervision, M.L. All authors have read and agreed to the published version of the manuscript.

**Funding:** The research of F.S. was accomplished within the project "UNICARagil" (FKZ 16EMO0288). We acknowledge the financial support for the project from the Federal Ministry of Education and Research of Germany (BMBF). The research of M.B. was conducted thanks to basic research funds from the Institute of Automotive Technology, Technical University of Munich.

**Institutional Review Board Statement:** Ethical review and approval were waived for this study, due to the voluntary questioning of experts about their knowledge without physical experimentation in creating the rule base of the fuzzy system.

**Informed Consent Statement:** Informed consent was obtained from all subjects involved in the study.

**Data Availability Statement:** The data presented in this study are available on request from the corresponding author. The data are not publicly available due to link to the respective expert.

**Acknowledgments:** M.L. gave final approval of the version to be published and agrees to all aspects of the work. As a guarantor, he accepts responsibility for the overall integrity of the paper. Many thanks to Philipp Hafemann for reviewing the paper.

**Conflicts of Interest:** The authors declare no conflict of interest.

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
