# Peer review of "Combining a Genetic Algorithm and a Fuzzy System to Optimize User Centricity in Autonomous Vehicle Concept Development"

_systems, doi:10.3390/systems9020025_

Round 1

Reviewer 1 Report

Review of the article: “Optimizing  user-centricity  in  autonomous  vehicle  concept  development combining a genetic algorithm and a fuzzy system”

Authors notice that future mobility solutions must satisfy more than just the need  for mobility.  They must offer the possibility of secondary activities and these user requirements must become part of the development process of AV concepts. The presented  in the article new methodology integrates user groups and their needs into the vehicle concept development  process. Thus, it is possible to design vehicle concepts for a private person using personas,  and to also optimize vehicle fleets for user groups. Regarding MaaS offerings with AV  fleets, this possibility is essential for user-oriented vehicle concept development.

General conclusion: The paper is well written and presents a good scientific level. The research topic is interesting and up to date. The proposed method is new.

The article includes:

- well-described and defined problem

- description of the proposed method with extensive discussion

- an example illustrating efficiency of the proposed algorithm.

- a discussion in which the strengths and weaknesses of the method were indicated

- future research directions.

The weaker side of the article is the introduction, which lacks a broader review of the literature and references to other studies in this field.

In the opinion of the reviewer, the article will be worth publishing after minor corrections.

Detailed comments:

line 709-710 - the same sentence twice

Reviewer 2 Report

The current version of the manuscript is well written and described. However, the authors can address the following points.

  1. The author should add more details in the introduction section.
  2. The author should rewrite the abstract and add more future implications in the conclusion.

Reviewer 3 Report

Driven by the megatrends of automation and sharing, authors have identified the demand for an evolution of customer-oriented concept development for a private individual using persons to a user-oriented approach using user group needs for AV concepts. For this purpose, authors present as a method a coupling of a genetic algorithm and a fuzzy system.

In general, authors present a concept combining the genetic algorithm and fuzzy system for optimizing user-centricity in autonomous vehicle. Authors should consider the following comments to clarify the main contributions of their paper.  

1.- In the page 1, in the introduction, authors say “This paper describes why we consider a combination of a genetic algorithm and a fuzzy system most suitable for this purpose and how we have implemented it.”, in this text, they should include references [a]-[f] which also consider the combination of genetic algorithm and fuzzy system.

[a] Novel Nonlinear Hypothesis for the Delta Parallel Robot Modeling, IEEE Access, Vol. 8, No. 1, pp. 46324-46334, 2020.

[b] SOFMLS: online self-organizing fuzzy modified least-squares network, IEEE Transactions on Fuzzy Systems, Vol. 17, No. 6, pp. 1296-1309, 2009.

[c] Wavelet-Based EEG Processing for Epilepsy Detection Using Fuzzy Entropy and Associative Petri Net, IEEE Access, Vol. 7, pp. 103255-103262, 2019. 

[d] Stability Analysis of the Modified Levenberg-Marquardt Algorithm for the Artificial Neural Network Training, IEEE Transactions on Neural Networks and Learning Systems, 2020. DOI: 10.1109/TNNLS.2020.3015200

[e] On the Estimation and Control of Nonlinear Systems With Parametric Uncertainties and Noisy Outputs, IEEE Access, Vol. 6, pp. 31968-31973, 2018.    

[f] Hybrid neural networks for big data classification, Neurocomputing, Vol. 390, pp. 327-340, 2020.

2.- In the pages 5, 6, in the section 3.2.2, authors should include equations to describe the genetic algorithm.  

3.- In the pages 6, 7, in the section 3.2.4, authors should include equations to describe the fuzzy system.  

4.- In the pages 5, 6, 7, in the sections 3.2.2 and 3.2.4, authors should include equations to describe how they combine the genetic algorithm and fuzzy system.  

5.- In the page 7, in the section 4, authors should clarify if they compare their method with other previous.

Round 2

Reviewer 3 Report

Authors attended the suggestions of the reviewer. Thus, this paper should be accepted as it is.